# Boosting the Initial Coulomb Efficiency of Sisal Fiber-Derived Carbon Anode for Sodium Ion Batteries by Microstructure Controlling

**DOI:** 10.3390/nano13050881

**Published:** 2023-02-26

**Authors:** Yuan Luo, Yaya Xu, Xuenuan Li, Kaiyou Zhang, Qi Pang, Aimiao Qin

**Affiliations:** 1Key Lab New Processing Technology for Nonferrous Metal and Materials, Ministry of Education, College of Material Science and Engineering, Guilin University of Technology, Guilin 541004, China; 2Guangxi Key Laboratory of Electrochemical Energy Materials, School of Chemistry and Chemical Engineering, Guangxi University, Nanning 530004, China

**Keywords:** bio-derived hard carbon, Coulomb efficiency, Na-ion batteries, storage mechanism

## Abstract

As anode material for sodium ion batteries (SIBs), biomass-derived hard carbon has attracted a great deal of attention from researchers because of its renewable nature and low cost. However, its application is greatly limited due to its low initial Coulomb efficiency (ICE). In this work, we employed a simple two-step method to prepare three different structures of hard carbon materials from sisal fibers and explored the structural effects on the ICE. It was determined that the obtained carbon material, with hollow and tubular structure (TSFC), exhibits the best electrochemical performance, with a high ICE of 76.7%, possessing a large layer spacing, a moderate specific surface area, and a hierarchical porous structure. In order to better understand the sodium storage behavior in this special structural material, exhaustive testing was performed. Combining the experimental and theoretical results, an “adsorption-intercalation” model for the sodium storage mechanism of the TSFC is proposed.

## 1. Introduction

Although lithium-ion batteries (LIBs), with a long cycle life and high energy density, are considered one of the most promising and successful energy storage systems today, they also face a number of problems, such as high cost and lack of resources. As a result, a new technology has emerged in the field of energy storage-sodium ion batteries (SIBs), which have electrochemical properties similar to LIBs [1,2,3]. Researchers have focused on the following types of anode materials: titanium-based oxides, sodium alloys, binary transition metal oxides (such as NiMoO_4_ [4] and MgMoO_4_ [5]), and carbon materials. Among the carbon-based materials that have been widely studied are graphite, graphene, soft carbon, and hard carbon. It is well known that graphite is the most common anode material in lithium-ion batteries, but its performance in sodium-ion batteries is not as expected because the Na^+^ radius (0.102 nm) is larger than the Li^+^ radius (0.076 nm) [6], Na^+^ cannot be stably embedded in the graphite structure, and only a very small number of binary sodium-graphite embedding compounds can be embedded in graphite [7]. Since the insertion and removal of large size sodium ions leads to a slowing down of the kinetic process, the structure undergoes irreversible phase changes, thus accelerating the degradation of the electrochemical properties, which implies a greater degree of limitation on the structure of the material [8,9,10].

Biomass hard carbon has attracted great attention due to its low cost, renewable nature, and green properties. Many biomass hard carbon materials, such as biowaste [11], eggshell [12], mango seed husk [13], etc., have been proven to have excellent storage performance when used in energy storage systems. However, most of them exhibit low initial Coulombic efficiency (ICE) for SIBs, i.e., 53.1%, 64.0%, and 69.0% of ICE for hard carbon from kelp [14], rice husk [15], and tea [16], respectively. Therefore, many researchers have discussed in depth the issues of defects, structural design, surface area, and conductivity to effectively improve the Coulomb efficiency. The properties of biomass char are mainly dependent on the nature of the raw material and thermal transformation. In order to make these biochar products more suitable for use as electrochemical energy storage materials, appropriate modifications are required to enhance the specific surface area, the pore volume, or the formation of functional groups. Usually, there are two methods for enhancement: physical or chemical. Physical activation usually involves a two-step process. Biomass materials are first pyrolyzed to generate biochar (400 °C to 850 °C) and then activated using gases such as CO_2_, air, or their mixtures [17]. These methods can reasonably design the structure of carbon material and improve the electrochemical properties. For example, Yan et al. [18] reported char balls obtained from nitrogen-rich oatmeal by hydrothermal and subsequent charring processes, exhibiting a smooth surface with an average diameter of about 2 μm; the results show that NCSs treated at 500 °C exhibit a high maximum charge capacity of 336 mAh g^−1^ after 50 cycles at a current density of 50 mA g^−1^. Duan et al. [19] prepared N-doped carbon microspheres by the pyrolysis of chitin from seafood waste (crab and shrimp shells), which consisted of nanofiber entanglements forming an interlinked nanofiber framework structure, and the highest deliverable energy density reached up to 58.7 Wh/kg. Jin et al. [20] prepared a series of porous hollow charcoal spheres using various spores (stone pine grass, Ganoderma lucidum, and multi-spike stone pine spores) as charcoal precursors and self-templates using high-temperature charring and activation treatments; the obtained electrodes showed remarkable electrical double-layer storage performances, such as high specific capacitance (308 F g^−1^ in organic electrolytes), ultrafast rate capability (retaining 263 F g^−1^ at a very high current density of 20 A g^−1^), and good cycling stability (93.8% retention after 10,000 charge-discharge cycles). Dong et al. [21] prepared a nitrogen-doped foamy charcoal plate by the charring and activation of teak peel and the prepared charcoal plate, with a macroporous network consisting of hollow tubes with diameters of 20–50 μm; the obtained electrodes showed a high specific capacitance of up to 338 F g^−1^ at 1 A g^−1^ and good rate capability with a capacitance retention of 59% at 20 A g^−1^.

In our group’s early efforts, we have conducted great work on the application of biomass carbon materials in LIBs. Among many studies, it was found that the hard carbon derived from cellulose-rich sisal fibers, with the advantages of low cost, green, and sustainable bioresource, exhibits excellent electrochemical performance when used as an anode for LIBs. The main focus of our previous work was to modulate the structure of sisal fibers using chemical activation and calcination methods, and to obtain sisal fiber carbon with various morphologies and structures [22,23,24] (its chemical activation reagents were KOH and HCl). Generally, the ICE of hard carbon can be effectively improved by controlling the defect concentration and specific surface area [25], but developing appropriate structures to ensure structural stability and high Coulombic efficiency, without affecting the electrolyte–electrolyte interface behavior, remains a challenge. In this work, in order to carry out an in-depth study on the relationship between the biomass hard carbon structures and their energy storage properties, for the first time, we chose sisal fibers as the study material for SIBs. We prepared three structure sisal fiber carbons: tubular sisal fiber carbon (TSFC), sheet sisal fiber carbon (SSFC), and spherical sisal fiber carbon (GSFC), using hydrothermal and calcination methods, and systematically investigated the different effects on ICE with the change of sisal fiber carbon structure. It was found that the special structure of TSFC can helps to improve electrochemical performance, especially greatly improving the ICE (76.7%), as well as the cycling stability. In addition, the diffusion kinetics and storage behavior of sodium ions in TSFC were further investigated.

## 2. Materials and Methods

### 2.1. Materials

All the chemical reagents used in the experiments were of analytical grade and were used without further purification. All aqueous solutions were prepared with deionized water. The sisal fiber used was from Guangxi Sisal Group, KOH reagent (Purity 85%, Guangdong Guanghua Chemical Factory Co., Ltd., Guangdong, China), and HCl reagent (AR 36–38%, Xilong Chemical Co., Shanghai, China).

### 2.2. Preparation of Tubular Sisal Fiber Carbon

The tubular sisal fiber carbon was prepared based on the methods used in our previous work [22], and the preparation process is as follows:

First, the sisal fiber was cleaned by washing, and 5 g of the clean, dry sisal fiber was placed into 2.5 mol L^−1^ KOH solution for hydrothermal reaction (160 °C, 14 h). When the reaction was finished and the autoclave was naturally cooled to room temperature, the precursor was collected after filtering and washing it to neutral. Then, it was dried in a blast drying oven (60 °C) for 24 h. Finally, the dried precursor was put into a crucible and calcined in a tube furnace at nitrogen atmosphere and kept at 900 °C for 1 h. The obtained black sample was collected in a crucible and named TSFC.

### 2.3. Preparation of Sheet Sisal Fiber Carbon

The sheet sisal fiber carbon was prepared based on the methods used in our previous work [23], and the preparation process is as follows:

First, the sisal fiber was cleaned by washing, and 5 g of the clean, dry sisal fiber was placed into 2.5 mol L^−1^ KOH solution for hydrothermal reaction (160 °C, 14 h). When the reaction finished and the autoclave was naturally cooled to room temperature, the precursor was collected after filtering and washing it to neutral. Then, it was dried in a blast drying oven (60 °C) for 24 h. A total of 3 g of the dried sample was weighed into a beaker, heated (80 °C), and stirred for 2 h (with 100 mL of 2.5 mol L^−1^ KOH solution), followed by drying. Finally, the dried precursor was put into a crucible and calcined in a tube furnace (nitrogen atmosphere, 900 °C, 1 h). The obtained black sample was collected in a crucible and named SSFC.

### 2.4. Preparation of Sphere Sisal Fiber Carbon

The sphere sisal fiber carbon was prepared based on the methods used in our previous work [24], the preparation process is as follows:

First, sisal fiber was cleaned by washing, and 5 g of the clean, dry sisal fiber was placed into a 2 mol L^−1^ HCl solution for hydrothermal reaction (180 °C, 12 h). When the reaction finished and the autoclave was naturally cooled to room temperature, the precursor was collected after filtering and washing it to neutral. Then, it was dried in a blast drying oven (60 °C) for 24 h. Finally, the dried precursor was put into a crucible and calcined in a tube furnace (nitrogen atmosphere, 900 °C, 1 h). The obtained black sample was collected in a crucible and named GSFC. In order to investigate the optimal conditions, we adjusted the concentration of HCl (0.1 M, 1 M, 4 M), which we named GSFC-0.1, GSFC-1, and GSFC-4, respectively.

### 2.5. Material Characterization

The morphology of the samples was observed by SEM (S4800) and TEM (JEM-2100F). The structure of the hard carbon material was characterized by XRD (X‘Pert PRO, PANalytical B.V., Cu Kα = 1.54056 Å, 2Θ = 10–80) and Raman ( Thermo Fisher Scientific DXR, Waltham, MA, USA), with the 532 nm excitation wavelength. The micropore volume and average pore size of the materials were characterized by nitrogen adsorption and desorption experiments (Quantachrome, Autosorb, Boynton Beach, FL, USA) at 77 k. The electrical conductivity of the material was measured by the four-probe method using model RTS-2A (Guangzhou Four-Probe Technology, Guangzhou, China) equipment.

The morphology of the samples was observed by SEM (S4800) and TEM (JEM-2100F). The structure of the hard carbon material was characterized by XRD (X‘Pert PRO, PANalytical B.V., Cu Kα = 1.54056 Å, 2Θ = 10–80) and Raman ( Thermo Fisher Scientific DXR, America), with the 532 nm excitation wavelength. The micropore volume and average pore size of the materials were characterized by nitrogen adsorption and desorption experiments (Quantachrome, Autosorb) at 77 k. The electrical conductivity of the material was measured by the four-probe method using model RTS-2A ( Guangzhou Four-Probe Technology, Guangzhou, China) equipment.

### 2.6. Electrochemical Measurements

The prepared active material was homogeneously mixed with conductive carbon black and polyvinylidene fluoride (PVDF) in the mass ratio of 8:1:1; then, N-methyl-2-pyrrolidone (NMP) was used as the solvent, and finally, the mixture was made into a paste of appropriate concentration. The electrolyte was 1.0 mol L^−1^ NaClO_4_ in ethylene carbonate (EC) and dimethyl carbonate (DEC), with a volume ratio of 1:1 and 5 vol% fluoroethylene carbonate. The electrochemical properties were tested using the CHI-760D electrochemical workstation. The voltage window for cyclic voltammetry (CV) measurement is 0.01–3 V and the electrochemical impedance spectra (EIS) were in the frequency range of 0.01–100,000 Hz. The Neware GCD testing system was utilized for galvanostatic charging/discharging profiles.

## 3. Results and Discussion

### 3.1. Morphology and Structure of Carbon Materials

The morphology and structure of the samples were represented by SEM and TEM. The SEM images of TSFC, SSFC, and GSFC at different magnifications are shown in Figure 1. From Figure 1a,b, it can be seen that the microstructure of TSFC consists of a hollow tube, with a wall thickness of ~1.4 µm and an inner tube diameter of ~4.5 µm. The special hollow tubular structure with a bumpy striped surface can promote the transfer of electrons. As shown in Figure 1c,d, the SSFC displays porous nanosheet structures, with a sheet thickness of about 10 nm and a pore size of 1–2 µm, and these interconnected nanosheets form a porous channel structure. Figure 1e,f presents the solid spherical structure of GSFC of about ~3 µm in size, with a smooth surface. In order to investigate the pretreatment condition effects on the morphology and structure of the materials, we take GSFC as an example to observe the evolution process of the spherical structure under different concentrations of HCl (0.1 M, 1 M, 2 M, and 4 M). The SEM results are shown in Appendix A. It can be seen that the morphology and structure of the material change significantly with the increase in the concentration of HCl: (i) in small concentrations of 0.1 M, small sphere particles grow on the surface of fibrous tubular carbon; (ii) when the concentration increases to 1 M, the elongated tubular fibers gradually disappear and become irregular spheres; (iii) when the concentration reaches 2 M, regular and more uniform spheres are obtained; and (iv) when the concentration continues to increase to 4 M, a large number of spheres are bonded together and the uniformity of the morphology decreases. We can assume that this structural change is due to the breakage of a large number of ester and ether bonds in the material during the degradation process [26]. Previously, the Austrian chemist Skrabal also mentioned that the decomposition of a compound is affected by a certain concentration of H^+^ or OH^−^ [26]. When cellulose hydrolyzes in an acidic environment, it produces a conjugate acid, which cleaves the β-1,4 glycosidic bond and interacts with H_2_O to produce H^+^, yielding more glucose [26]. Glucose is cleaved to form a large number of carboxylic acids, aldehydes, and furans, which are dehydrated and condensed to form aromatic compounds, which in turn form spherical carbon materials [26,27]. On the contrary, in an alkaline environment [26], cellulose will form a large number of organic acids during hydrolysis to neutralize the alkaline substances of the reaction system, which will cause a large amount of H^+^ to be consumed, resulting in inadequate cellulose dissolution and stripping the cellulose away. The formation of a lamellar porous structure is mainly due to the reaction of KOH and C at high temperatures to produce H_2_, CO_2_, and CO, which promote the formation of porous carbon materials [28].

As depicted in Figure 2a–f, the TEM diagram clearly exhibits the microstructure of the materials. Figure 2b,d,f shows the high resolution transmission microscopy (HRTEM) images of TSFC, SSFC, and GSFC, respectively. We can clearly see that the crystal plane spacing of all three structures is larger than the graphite layer spacing (graphite layer spacing is 0.335 nm), and the crystal plane spacing of TSFC (0.37 nm) is significantly larger than the other two morphologies. It has also been demonstrated in the literature that Na ions cannot be inserted into graphite with a layer spacing smaller than 0.335 nm, but they can be easily inserted into graphite with a layer spacing of 0.37 nm [29], so we can expect that the TSFC anode will exhibit superior sodium storage performance compared to the other two structures.

As shown in Figure 3a,b, the microstructure of samples was analyzed by XRD and Raman spectroscopy. The XRD patterns of TSFC, SSFC, and GSFC display two main broad peaks at about ~23° and 43° corresponding to (002) and (101) lattice planes, respectively [30], demonstrating the amorphous state of the samples. A shift in the (002) peak toward a small angle can be observed for the TSFC material, indicating that the interlayer spacing of the structure also changes to some extent. This is consistent with the TEM results, indicating that the layer spacing of biogenic hard carbon changes with the change of morphology [31]. Additionally, the disordered and graphitized structures of samples have been assessed by Raman spectroscopy. The Raman spectra show two separate characteristic bands of the D-band peak at ~1330 cm^−1^ and the G-band peak at 1580 cm^−1^, corresponding to the D-band, with sp3 defects, and the G-band, with ordered graphite sp2 features [32]. The value of I_D_/I_G_ is often used to characterize the defects in carbon materials [33]. It can be seen that the I_D_/I_G_ value of TSFC sample (0.852) is higher than that of the SSFC (0.847) and GSFC (0.835) samples, indicating that TSFC has more defects and may provide more active sites for sodium storage [34]. The porosity of the samples was further investigated by N_2_ adsorption-desorption tests. The N_2_ adsorption-desorption isotherms of TSFC, SSFC, and GSFC are shown in Figure 3c, where we can observe that the three structures exhibit the type IV isotherms. For TSFC and SSFC, their isotherms show a sharp rise at relative pressures of less than 0.1, followed by bending into a platform accompanying the H4 reversible hysteresis loop, implying the presence of well-developed microporous and mesoporous structures. The average pore size and pore volume were calculated by the Barrett–Joyner–Halenda (BJH) method and the single point method. The pore size distribution of the three structures in Figure 3d indicates the presence of well-developed mesoporous structures in the three hard carbon structure materials [32]. The porous structure details are given in Appendix A, and compared with other two samples, TSFC has the most developed porous structure, with an average pore diameter of about 2.79 nm and a medium specific surface area of 426.02 m^2^ g^−1^, as well as a moderate open pore volume of 0.049 cm^3^ g^−1^. However, it is worth noting that the increase in microporous volume leads not only to a low density of the hard carbon material, but also to a decrease in intercalation capacity due to the reduction of graphite-like nanodomains in the hard carbon structure [35]. Therefore, combining all the above data references, the hollow tubular structure is favorable for soaking the electrolyte and will promote the ion transport from electrolyte to electrode.

### 3.2. Electrochemical Performances

To further investigate the effect of structure on the sodium ion storage properties, we performed cyclic voltammetry CV tests for the first three turns at a scan rate of 0.5 mV s^−1^. As shown in Figure 4a–c, it can be observed that the CV curves of the first three turns of the three electrode materials are very similar, with a pair of redox peaks at around 0.01 V/0.16 V, corresponding to the insertion and extraction process of Na^+^. In the initial cathodic scan, the three electrodes show irreversible reduction peaks, with different intensities. This may be related to the irreversible reactive binding of Na^+^ on the surface functional groups or other defective sites and the formation of SEI films in the first cycle [36]. Compared to SSFC and GSFC, TSFC shows a large sharp peak below 0.1 V, demonstrating an enhanced sodium storage [37]. All the CV curves for the TSFC (Figure 4a) have an excellent overlap during the subsequent cycles, indicating that the electrode material has good reversible capacity and cycle stability. The EIS plots and four-probe resistance tests for the three electrodes are shown in Figure 4d,e, respectively. The EIS plots are composed of a semicircle at high frequency and a slope line at low frequency, which correspond to the interaction of Na^+^ with the SEI layer/charge transfer between the electrolyte and the active material and the diffusion of Na^+^ in the active material, respectively [38]. It is clear that the TSFC anode exhibits a lower Rct (32.8 Ω) than do SSFC and GSFC, which can be attributed to its large aperture nanochannels, increasing surface wettability and making the electrolyte flow easy throughout the electrode [39]. To further demonstrate the electrical conductivity of the material, we tested the positive and negative currents of the electrode sheet using a four-probe resistivity tester (Figure 4e). The test results show that the TSFC electrode material possesses the lowest conductivity (2.49 Ω·cm), indicating that the structure plays an important role in the resistance of the material and helps to improve the sodium ion transport, which corroborates with the EIS results. Figure 4f shows the discharge curves of TSFC, SSFC, and GSFC at a current density of 0.1 A g^−1^. All discharge curves show two areas: the sloping portion (3.0 V~0.1 V) and the plateau portion (<0.1 V). The initial charge/discharge capacities of SSFC, GSFC, and TSFC are 131.11/461.6, 170.8/330.8, and 265.2/345.9 mAh g^−1^, respectively. The ICE of the three structures also varies from 28.4% to 76.7%, depending on their structures. It is obvious that TSFC has a better first charge/discharge efficiency than the other two samples. The reason is that a proper specific surface area will reduce the occurrence of side reactions, thus reducing the irreversible capacity and thus exhibiting a high initial Coulomb efficiency [40]. The discharge curves of TSFC, SSFC, and GSFC exhibit sloping capacities of 216 mAh g^−1^, 340 mAh g^−1^, and 265 mAh g^−1^, and plateau capacities of 132 mAh g^−1^, 120 mAh g^−1^, and 66 mAh g^−1^, respectively. Obviously, the capacities of the three anodes are mainly derived from the adsorption of Na^+^ (sloping part). The plateau contributions of TSFC, SSFC, and GSFC are 38%, 26%, and 20% of the total capacity, respectively. There is no doubt that the plateau part of TSFC contributes more capacity than that of the other two, which is mainly attributed to the larger Na^+^ intercalation layer provided by the layer spacing of TSFC. Cyclic stability over 100 cycles was recorded for the samples at 100 mA g^−1^, as shown in Figure 4g. Compared to SSFC and GSFC, TSFC performed relatively well, with an initial specific capacity of 265.2 mAh g^−1^ and an ICE of 76.7%, while the ICE of SSFC and GSFC were only 28.4% and 51.6%. This indicates that the specific surface area has a greater effect on ICE, which has been confirmed in the literature [41]. Appendix A provides a comparison of the electrochemical properties of various previously reported biomass-derived carbon materials with the materials prepared in this work. It is clear that the TSFC anode material possesses a higher ICE. In addition, the TSFC also has a higher ICE than that of some previously reported transition metal materials (such as NiMoO_4_ [4] and MgMoO_4_ [5]). For the stability of the electrode material structure, multiplicative performance tests were performed on three electrodes (as shown in Figure 4h). At specific current densities of 0.02, 0.05, 0.1, 0.5, 1.0, and 2.0 A g^−1^, the specific capacities of TSFC are 281.2, 257.7, 235.0, 103.6, 70.14, and 24.6 mAh g^−1^, respectively. When the current density returns to 0.02 A g^−1^, a capacity of 255.6 mAh g^−1^ is obtained, demonstrating a good rate performance and the capacity retention rate remained at 90.8%. On the contrary, an inferior rate performance of the SSFC and GSFC is observed under the same condition. The long-cycle performance of the TSFC electrode is also evaluated, as shown in Figure 4i. The capacity of TSFC could be maintained at 110 mAh g^−1^ after 400 cycles at 50 mA g^−1^, and the charge/discharge efficiency was always maintained at about 100%, indicating its excellent durability and potential.

In order to explain the excellent sodium storage performance of TSFC, the CV curve at different scanning rates of 0.2, 0.4, 0.6, 0.8, and 1 mV s^−1^ (vs. Na/Na^+^) was measured, as shown in Figure 5a, in which the CV curves exhibit a certain deviation from the rectangle, demonstrating the combination of two different charge storage mechanisms of faradaic and non-faradaic reactions [42]. As the scan rate increases, the shape of the CV curve remains constant, without serious distortion, indicating its high reversibility and excellent multiplicative performance [42]. Generally, the relationship between peak current (*i*) and scan rate (*v*) obeys Equation (1) [43]:(1)log(i)=log(a)+blog(v)

Here, *b* represents the slope of log(*v*) and log(*i*), where *b* values were in the range of 0.5 to 1, indicating a pseudo-capacitance contribution, in addition to diffusion-controlled intercalation behavior at these potentials. The calculation result of *b* values can be seen in Figure 5b. The b-value of TSFC is 0.75, indicating that it has a hybrid mechanism of pseudo-capacitance contribution and diffusion-controlled intercalation behavior. Furthermore, the percentage contribution of the capacitive process in the electrochemical reaction process can be calculated according to Equation (2) [43]:(2)$$i\left(v\right)=k_{1}v+k_{2}v^{\frac{1}{2}}$$
where *k*_1_*v* is the surface capacitive contribution, and *k_2_v*^1/2^ is the diffusion contribution. The calculated results are shown in Figure 5c,d, where it can be observed that the pseudocapacitance contribution takes up an increasing percentage as the scan speed increases. This phenomenon is probably due to the special hierarchical porous structure of TSFC, which induces the shortened transport pathway, faster delocalization, and the embedding of sodium ions. The pseudocapacitance tests were also analyzed for SSFC and GSFC under the same conditions, and the detailed data are shown in Appendix A. It can be seen that the b-value of the SSFC and GSFC electrodes is 0.57 and 0.82, respectively. Thus, a hybrid mechanism exists in the three electrode materials. The pseudocapacitance percentage increases as the scanning speeds increase, which are 32%, 39%, 45%, 48%, and 51% for TSFC; 46%, 53%, 57%, 63%, and 65% for SSFC; and 44%, 52%, 58%, 61%, and 64% for GSFC. However, the contribution of the diffusion-controlled interpolation behavior of TSFC is higher than that of the other two at different sweep speeds, and this result is exactly consistent with the above analysis results in Figure 4f.

Figure 6a–f show the GITT reaction curves of TSFC, SSFC, and GSFC; the Na^+^ diffusion coefficients of the three electrode materials fluctuate between 10^−2^ cm^2^ s^−1^ and 10^−6^ cm^2^ s^−1^, but the Na^+^ diffusion coefficient of TSFC has small fluctuation at the beginning of discharge, indicating that an enhanced sodium diffusion occurred during sodium adsorption [44]. As the voltage slowly decreased, the diffusion of sodium ions gradually smoothed out. When the voltage dropped down below 0.1 V, the sodium ion diffusion coefficient decayed rapidly. We believe that the diffusion of D_Na_^+^ is caused by the change in the sodium ion storage mechanism from the adsorption to insertion type [45]. Interestingly, the D_Na_^+^ of TSFC tends to rise at the end of the discharge process. This phenomenon is common in the high-temperature treated hard carbon materials, suggesting that TSFC can exhibit similar structural properties after electrochemical reactions [45]. In addition, when using TSFC as an electrode material, the charging and discharging duration of TSFC is up to 73 h, while the charging and discharging duration of SSFC and GSFC is only 39 h and 68 h, respectively, which is clearly longer than that of the other two samples. This result is also consistent with the results of the above comprehensive analysis.

To investigate the sodium storage mechanism of the TSFC anode materials in detail, we recorded the capacity variation of the TSFC discharge curve from the 1st cycle to the 400th cycle at a current density of 0.05 mA g^−1^; the direct effect on the low voltage plateau capacity and high voltage sloping capacity was also explored in detail. As shown in Figure 7, the sloping capacity of TSFC increases from 62% in the 1st cycle to 82.5% in the 400th. The curves show that the contribution of ramp capacity predominates during high voltage discharge, indicating that in TSFC, the special hierarchical porous structure provides more active sites for the physical/chemical adsorption of Na^+^. As the number of cycles increases, the number of electrochemical reactions increases accordingly from the reactions that initially occur on the electrode surface, slowly deepening to the interior, while the micropores open at this time with the successive sodiation/desodiation cycle reactions, at which time the electrolyte enters the micropores more easily, leading to an increase in sloping capacity [46]. Moreover, the combined contribution of intercalation and micropore filling to the plateau capacity is believed to play a role in the overall sodium storage process when the voltage is gradually reduced to 0.1 V. According to the literature, a layer spacing of 0.36–0.40 nm is required for the viability of sodium ion insertion/extraction from amorphous carbon in the low-voltage plateau region [47]; the layer spacing of TSFC is 0.37 nm, which is right in this range. On the other hand, hard carbon materials possess a complex structure where porosity and layer spacing are the main objective conditions for sodium storage. The presence of inflection points at the end of the potential of the discharge curve also becomes the basis for determining the predominance of interlayer intercalation and microporous filling [35]. As for the low-voltage plateau region of TSFC, there was no electrochemical inflection point when the discharge gradually reached 0 V. Therefore, based on the above analysis we reasonably believe that the sodium storage mechanism of TSFC is “adsorption-intercalation/filling,” and the model is shown in Figure 7.

Based on the above analysis, it can be concluded that the TSFC anode exhibits excellent ICE and multiplicative performance, which may be attributed to the following characteristics: (i) the specific surface area can effectively reduce the generation of side reactions and further improve the Coulomb efficiency; (ii) the larger layer spacing can provide efficient ion channels and accelerate the Na^+^ transport; and (iii) the specific hierarchical porous structure facilitates the rapid diffusion of electrolyte ions into the interior of the electrode material, providing more active surface sites and thus increasing the slope capacity contribution.

## 4. Conclusions

In conclusion, three different structure hard carbon materials (TSFC, SSFC, and GSFC) were prepared via a low-cost and simple method to treat sisal fibers through varying the pretreatment conditions under acid or alkali systems, and a detailed comparative study on the morphology and the electrochemical properties of these structures was conducted. The effect of biomass hard carbon structures on the initial Coulomb efficiency was carefully explored. It was found that the tubular structure of sisal fiber carbon (TSFC) shows a medium specific surface area, larger layer spacing, and a special hierarchical porous structure, which favors efficient ion transport channels, ion migration, and storage. The reversible specific capacity of TSFC can maintain 110 mAh g^−1^ at a current density of 0.05 A g^−1^ for 400 cycles, and the charge/discharge efficiency was maintained at about 100%. TSFC also exhibits a great increase in ICE (76.7%), along with cycling stability, compared with SSFC and GSFC. To investigate the effect of structural changes on its sliding and plateau capacities, the discharge curves of the TSFC with different numbers of cycles were analyzed, and the “adsorption-intercalation/filling” model for the sodium storage mechanism of TSFC was proposed; that is, at low voltage, the plateau capacity is related to the intercalation/filling of sodium ions in materials, and the slope capacity at high voltage is related to the adsorption behavior of the sodium ions. In summary, this study provides a new approach for obtaining high ICE for sodium ion battery anodes.

## Figures and Tables

**Figure 1 nanomaterials-13-00881-f001:**
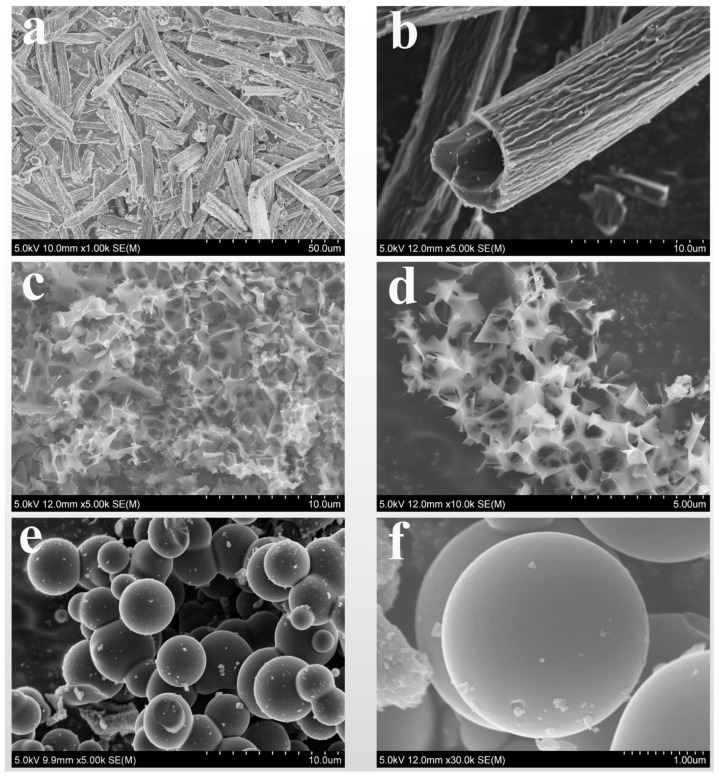
The different magnification SEM images of (**a**,**b**) TSFC, (**c**,**d**) SSFC, (**e**,**f**) GSFC.

**Figure 2 nanomaterials-13-00881-f002:**
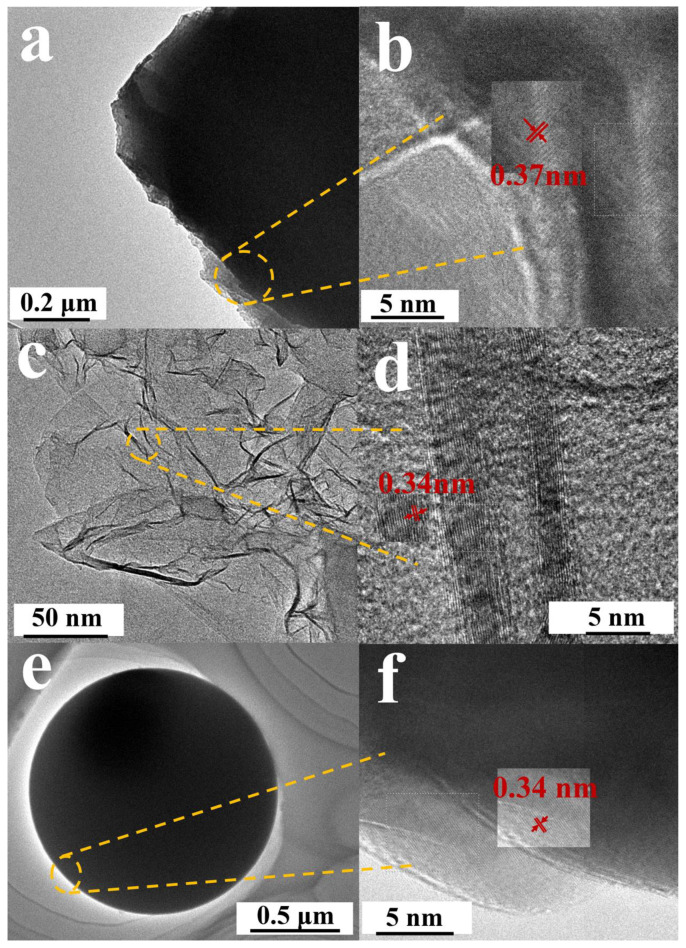
TEM images of (**a**,**b**) TSFC, (**c**,**d**) SSFC, (**e**,**f**) GSFC.

**Figure 3 nanomaterials-13-00881-f003:**
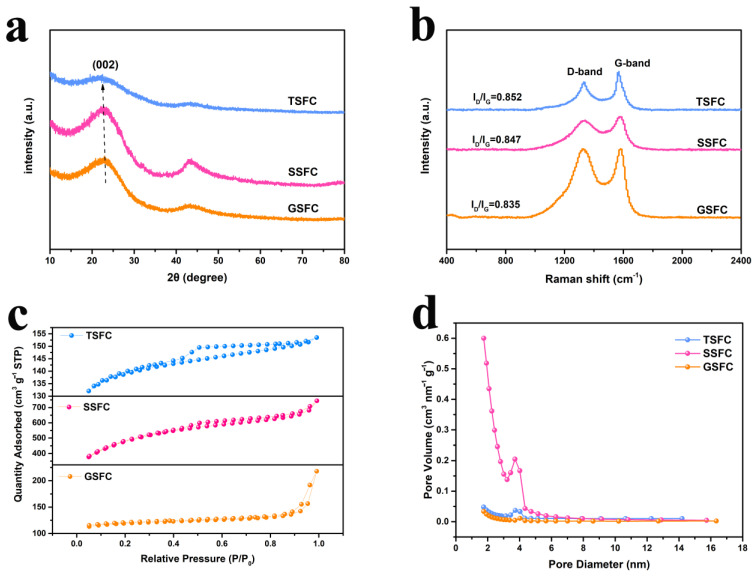
Structural characterization of TSFC, SSFC, and GSFC: (**a**) XRD patterns; (**b**) Raman spectrum; (**c**) N_2_ adsorption/desorption isotherms; (**d**) pore size distributions.

**Figure 4 nanomaterials-13-00881-f004:**
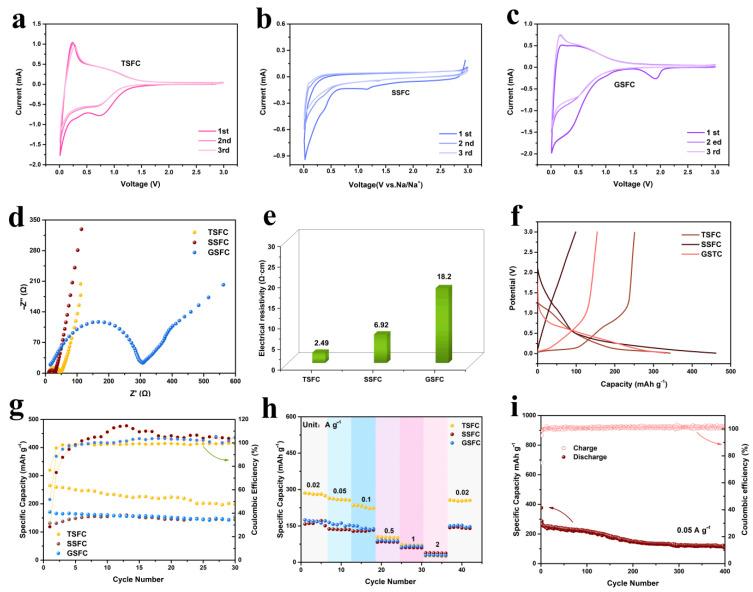
(**a**–**c**) CV curves of the first three circles, (**d**) EIS spectra, and (**e**) the electronic conductivity of the TSFC, SSFC, and GSFC anodes. (**f**) The initial galvanostatic discharge-charge curves at 0.1 A g^−1^; (**g**) comparison of the three anodes cycling performance at 0.1 A g^−1^; (**h**) rate performance of the three anodes; (**i**) prolonged cycling performance of the TSFC anode at 0.05 mA g^−1^.

**Figure 5 nanomaterials-13-00881-f005:**
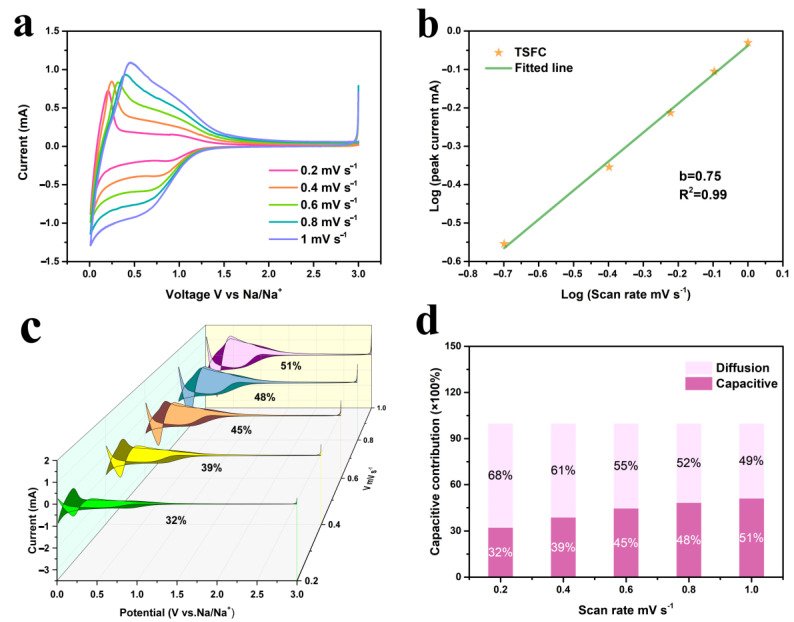
(**a**) Voltammetry curves of TSFC at different scan rates between 0.2 and 1 mV s^−1^; (**b**) the plots of log(*i*) versus log(*v*) of TSFC; (**c**) pseudocapacitive contribution waterfall diagram of TSFC; (**d**) pseudocapacitive contribution of TSFC at different scan rates between 0.2 and 1 mV s^−1^.

**Figure 6 nanomaterials-13-00881-f006:**
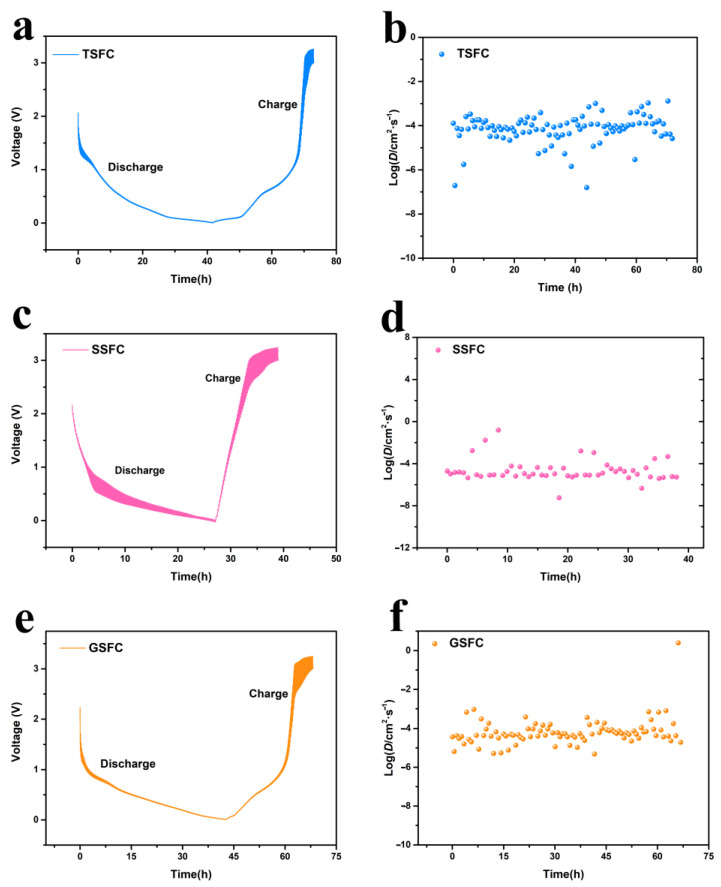
(**a**–**f**) Shows the GITT reaction curves of the composite of TSFC, SSFC, and GSFC.

**Figure 7 nanomaterials-13-00881-f007:**
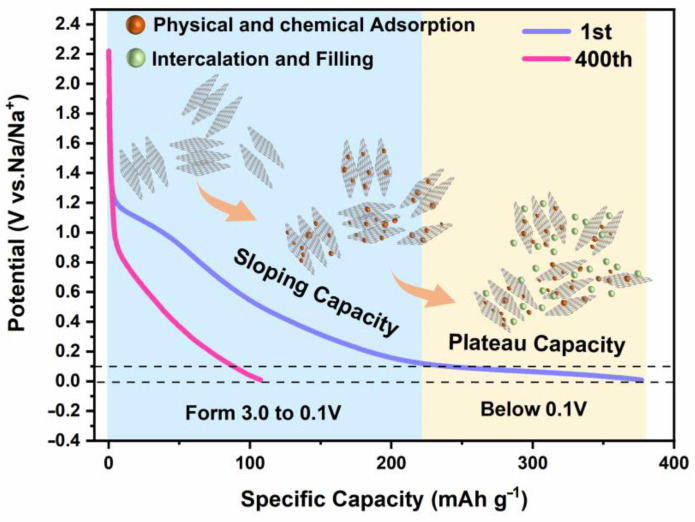
Sodium−ion storage mechanism of TSFC.

## Data Availability

Data available within the article or its Appendix A.

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
