# Peer review of "Boosting the Initial Coulomb Efficiency of Sisal Fiber-Derived Carbon Anode for Sodium Ion Batteries by Microstructure Controlling"

_nanomaterials, 2023, doi:10.3390/nano13050881_

Round 1
Reviewer 1 Report
An article entitled "Boosting the Initial Coulomb Efficiency of Sisal Fiber-Derived Carbon Anode for Sodium Ion Batteries by Microstructure Controlling" by Yuan Luo, Ya Ya Xu, Kai You Zhang, Qi Pang, Ai Miao Qin is a well written research paper that could be published in Nanomaterials. However there are some points, that should be taken into account before publication:
- Materials description in section 2.1 is insufficient. Authors need to present a more detailed description of the reagents, their manufacturer, purity, etc.
- Methods of preparation (sections 2.2, 2.3, 2.4) could be also improved. Were these methods invented by the authors or taken from literary sources? Why were such conditions for temperature and time chosen?
- Careful check of the text is needed, superscript and subscript characters, extra or missing spaces and punctuation marks, etc.
- Conclusions should be extended
Reviewer 2 Report
This manuscript describes the preparation of three different structures of biomass-derived hard carbon materials from sisal fibers. The morphology and electrochemical performance of this structure were entirely investigated and compared. The effect of hard carbon structures on initial Coulomb efficiency was carefully explored. In my opinion, this manuscript needs to have minor revisions to be published in this journal, following reasons:
1. In my opinion, the reason for choosing sisal fibers to become the study material should be mentioned in the introduction.
2. Chemical reagent information should be written in more detail.
3. The hollow carbon structure drawn in the graphic abstract should be adjusted to correspond with the one in SEM.
4. The TEM images should be reorganized and annotated clearly.
5. Based on the SEM and TEM images of the sphere carbon structure (GSFC), no porous could be observed. Could you explain the average pore diameter value and the open pore volume value of this structure?
In my opinion, this manuscript should be submitted after minor revision. Therefore, I recommend accepting this manuscript for publication in this journal.
Reviewer 3 Report
Lithium-ion batteries are now dominating due to their excellent electrochemical performance such as exhibiting high energy density, long cycling life, minimum self-discharge, and limited memory effect. The foremost advantage of Na-ion batteries comes from the natural abundance and lower cost of sodium compared with lithium. We can foresee Na-ion batteries with hard-carbon anodes and cobalt-free cathodes as sustainable lower-cost alternatives to Li-ion batteries for applications such as short-range electric vehicles. This is a significant area of research. Graphite is traditionally used as an anode and offers satisfactory capacity and stable cycling life despite the issues of the formation of lithium dendrites. Alternatively, titanium dioxide (TiO2) has been considered one of the potential electrodes due to its excellent stability. This is a well-researched material; however, several critical drawbacks have significantly limited their practical applications. In the submitted work, the authors have used sisal a kind of cheap renewable biomass resource, which has excellent mechanical properties are mainly used as textiles, artware, and reinforced material. The paper itself is written OK but the main concern of this paper and the areas that require clarification are given below.
The following points need to be considered.
· In the title, it says “microstructure controlling” but in the introduction it says nothing.
· The last paragraph of the introduction is dubious; what is a two-step method? What are three biomass-derived carbon materials? How are these expected to influence ICE? Importantly, the objectives of the paper and what is new in the submitted work must be demonstrated in the introduction section.
· The importance of TSFC has not been addressed well throughout the paper.
· Sisal fiber has been reported earlier in the domain of anode materials (such as doi.org/10.1016/j.matlet.2014.11.160; and by the authors themselves Int. J. Electrochem. Sci., Vol. 11, 2016). Being this case, the level of originality must be emphasized clearly otherwise the importance of the work would be misleading. Gives an impression as though sisal fiber itself is new but that is not the case.
· The key papers on carbon anode reported for batteries derived from renewable carbon resources (such as biowaste, mango seed husk, eggshell, grape marc) reported by Manickam Minakshi et al can be included while bringing out the pros and cons of those over sisal fiber.
· Lines 30 – 31 Researchers have focused on three main types of anode materials: Titanium-based oxides, sodium-alloy, and carbon materials” true but also other binary transition metal oxides as anodes for sodium-ion batteries such as MgMoO4, NiMoO4, and CaMoO4, etc. Please refer to and include.
· Line 130 – fix the syntax error.
· The clarity of the figures can be improved.
· The shape and the CV characteristic observed in Fig. 5a must be explained well with the mechanism involved.
· Was there a formation of SEI that reduces the initial discharge capacity after 400 cycles?
Please also benchmark the storage performance with the reported anode material both in binary transition metal oxides and other hard carbons.
Reviewer 4 Report
The author describes “Boosting the Initial Coulomb Efficiency of Sisal Fiber-Derived Carbon Anode for Sodium Ion Batteries by Microstructure Controlling”. This paper is quite interesting from a technological point of view. The author should revise their manuscript based on the comments and suggestions. I recommended a Major revision of the manuscript.
The Major suggestion below:
- The author should provide the different magnification SEM images of TSFC, SSFC, and GSFC.
- The author should provide the EDX spectra of TSFC, SSFC, and GSFC.
- From the TEM, I could not find any Tube-like structure for TSFC. Please provide the different TEM images of TSFC.
- In BET analysis, all three samples showed three different types of the hysteresis loop, but the author mentioned only the type IV and H4 loop only why? They should explain in detail.
- In BET (fig.3 c and d), the author should use the same color for both data.
- The figure quality is too poor and the author should improve the quality of the images. As all figure formats are different, the author should check clearly.
- The author discussed the ICE, so the author should provide the after-electrochemical test SEM and XRD for better understanding.
Round 2
Reviewer 3 Report
I went through the revised part of the manuscript and the responses made by the authors. The revised version is suitable for publication.
Reviewer 4 Report
The authors carried out all my comments. So I recommend for acceptance.